# Bovine tuberculosis in African buffalo (*Syncerus caffer*): Progression of pathology during infection

**Hilary Ann Lakin**[1]*, **Hannah Tavalire**[2], **Kaori Sakamoto**[3], **Peter Buss**[4], **Michele Miller**[5], **Sarah A. Budischak**[6], **Kristina Raum**[1], **Vanessa O. Ezenwa**[7], **Brianna Beechler**[1‡], **Anna Jolles**[1,8‡]

**1** Carlson College of Veterinary Medicine, Oregon State University, Corvallis, Oregon, United States of America, **2** Prevention Science Institute, University of Oregon, Eugene, Oregon, United States of America, **3** Department of Pathology, College of Veterinary Medicine, University of Georgia, Athens, Georgia, United States of America, **4** Veterinary Wildlife Services, South African National Parks, Skukuza, South Africa, **5** DSI/NRF Centre of Excellence for Biomedical Research/MRC Centre for Tuberculosis Research, Division of Molecular Biology and Human Genetics, Faculty of Medicine and Health Sciences, Stellenbosch University, Cape Town, South Africa, **6** W.M Keck Science Department, Claremont McKenna, Pitzer and Scripps Colleges, Claremont, California, United States of America, **7** Department of Ecology and Evolutionary Biology, Yale University, New Haven, Connecticut, United States of America, **8** Department of Integrative Biology, Oregon State University, Corvallis, Oregon, United States of America

‡ These authors are joint senior authors on this work.
* lakinh@oregonstate.edu

**Data Availability Statement:** All data is contained within the public repository, accessible at https://zenodo.org/record/4944360#.Y17d_ezMJ9s, https://datadryad.org/stash/dataset/doi:10.5061/dryad.pk0p2ngmh.

## Abstract

### Background

Bovine tuberculosis (BTB) is a zoonotic disease of global importance endemic in African buffalo (*Syncerus caffer*) in sub-Saharan Africa. Zoonotic tuberculosis is a disease of global importance, accounting for over 12,000 deaths annually. Cattle affected with BTB have been proposed as a model for the study of human tuberculosis, more closely resembling the localization and progression of lesions in controlled studies than murine models. If disease in African buffalo progresses similarly to experimentally infected cattle, they may serve as a model, both for human tuberculosis and cattle BTB, in a natural environment.

### Methodology/Principal findings

We utilized a herd of African buffalo that were captured, fitted with radio collars, and tested for BTB twice annually during a 4-year-cohort study. At the end of the project, BTB positive buffalo were culled, and necropsies performed. Here we describe the pathologic progression of BTB over time in African buffalo, utilizing gross and histological methods. We found that BTB in buffalo follows a pattern of infection like that seen in experimental studies of cattle. BTB localizes to the lymph nodes of the respiratory tract first, beginning with the retropharyngeal and tracheobronchial lymph nodes, gradually increasing in lymph nodes affected over time. At 36 months, rate of spread to additional lymph nodes sharply increases. The lung lesions follow a similar pattern, progressing slowly, then accelerating their progression at 36 months post infection. Lastly, a genetic marker that correlated to risk of *M. bovis* infection in previous studies was marginally associated with BTB progression.

**Funding:** Funding for captures and diagnostics was provided by the National Science Foundation (DEB-1102493/EF-0723928 andEF-0723918 to VO and AJ). Funding for molecular genetic work was provided by Morris Animal Foundation (D15ZO-824) to AJ and HT. Funding for HAL was provided by the Oregon State University, Carlson College of Veterinary Medicine, Biomedical Sciences program. The funders had no role in study design, data collection and analysis, decision to publish or preparation of the manuscript.

**Competing interests:** The authors have declared that no competing interests exist.

Buffalo with at least one risk allele at this locus tended to progress faster, with more lung necrosis.

## Conclusions/Significance

The progression of disease in the African buffalo mirrors the progression found in experimental cattle models, offering insight into BTB and the interaction with its host in the context of naturally varying environments, host, and pathogen populations.

### Author summary

Bovine tuberculosis affects many mammals worldwide, including the African buffalo. Within the African buffalo population in Kruger National Park, South Africa, bovine tuberculosis is endemic, thus buffalo within this area are regularly infected and act as a reservoir for infection of other wildlife, livestock, and humans in the area. Due to the risk to humans and other mammals, bovine tuberculosis is considered a disease of global importance; cattle are used to model the disease progression in humans. This study seeks to compare the progression of bovine tuberculosis in free-ranging buffalo to the progression in experimental cattle models. Free-ranging buffalo encounter more variables than experimental cattle, including variations in feed, co-morbidities, and birthrate, similar to humans. Overall, the progression of disease, both grossly and microscopically, in African buffalo mirrors the progression in cattle models despite the increase in variables, providing support that free-ranging models can be used for disease progression studies, with the added benefit of representing the variation in lifestyles present outside of controlled studies. Lastly, we looked at the genetic basis of disease within the herd and found a genetic marker that marginally correlated to disease progression, indicating a need for further work understanding the genetic basis of bovine tuberculosis.

## Introduction

*Mycobacterium bovis* infection, resulting in zoonotic tuberculosis in humans and bovine tuberculosis (BTB) in animals, is currently endemic in many African countries, with the African buffalo (*Syncerus caffer*) serving as one of the maintenance hosts [1]. In endemic areas, infection can spill over to cattle, other domestic animals, vulnerable wildlife species and humans [2,3]. In countries where BTB is endemic in domestic bovids (primarily in Africa and Southeast Asia, *M. bovis* results not only in potential zoonotic sources of infection but has a major impact on economics and trade barriers of countries where BTB is endemic, primarily in Africa and Southeast Asia [4,5]. Due to a lack of routine surveillance data, the burden of zoonotic TB is most likely underestimated; however, in 2016 there were an estimated 147,000 new cases of zoonotic TB in people, and 12,500 deaths due to the disease globally [4,5]. High prevalence of *M. bovis* in the cattle population is associated with the highest reporting of zoonotic tuberculosis in humans [6]. The ability to identify key wildlife reservoir populations and reduce the prevalence of TB in livestock is pivotal and understanding pathogenesis of disease forms the basis of a global health strategy [4,7].

Understanding the pathogenesis of BTB in wild and domestic bovid populations is not only valuable in the context of zoonotic disease, but in supplementing our understanding of human tuberculosis, caused by the closely related *Mycobacterium tuberculosis*. The progression pattern

of BTB in controlled large animal studies has been proposed as a model of infection for human tuberculosis due to clinical and immunological similarities [8]. Cattle may also be a better model when compared to rhesus macaque studies due to the lower cost, ethical advantages, and lower health risks for animal care takers. As with *M. tuberculosis*, laboratory studies in cattle have shown *M. bovis* can be acquired through several routes in bovid species [3], most commonly by inhalation [2]. The acid-fast bacilli are inhaled and reach the alveoli, where pulmonary alveolar macrophages phagocytose the bacteria [9]. As a facultative pathogen of the monocytic-macrophage system, *M. bovis* multiplies intracellularly within the macrophage, eventually killing the cell and spreading infection within the host [2]. The bacilli then spread aerogenously via airways within the lungs and lymphatically to the lymph nodes [2].

In cattle, the early stage of BTB infection occurs primarily in the lymph nodes of the head and thoracic cavity, usually detected by visual inspection for lesions at necropsy or bacterial culture [8]. The majority of these early lesions are found in the retropharyngeal lymph nodes, draining the tonsils, and mediastinal lymph nodes [10]. During intermediate stages of BTB, one or more visible lesions become apparent in the lungs. In late stages, BTB infection is characterized by a granulomatous pneumonia. The resulting multifocal, nodular lesions visibly range in appearance from pyogranulomatous, caseous necrosis, to calcified nodules [2].

Microscopically, in cattle and humans, pulmonary granulomas caused by *Mycobacterium tuberculosis* complex bacteria (commonly referred to as tubercles), appear as a center of necrotic tissue, surrounded by a rim of macrophages and giant cells. Commonly, an outer layer of connective tissue infiltrated by lymphocytes and plasma cells delineates the tubercle from the normal lung tissue [2]. Conversely, tuberculosis in lab animals, such as guinea pigs, rabbits, and mice, has a progression independent from bacterial load that is often evident in the spleen and liver, unlike progression in bovids and humans [8]. Furthermore, in cattle, like humans, the organization of the granulomas, is heterogenous, with primary foci that infect other parts of the lungs, commonly with necrosis [6,8,11]. In mice, the granuloma is homogenous, rarely shows necrosis, and follows a less organized progression from formation through necrosis to liquefaction. The guinea pig and rabbit often die before the cavities become larger or reach liquification, in part due to the seeding to the spleen and liver [6]. Studies completed in free-ranging African buffalo show similar common sites of gross lesions and microscopic tubercle formation as cattle and humans [12]. Our study proposes that not only cattle, but other free-ranging bovids may serve as a model for human tuberculosis, broadening the potential candidates for animal models of disease.

Bovine TB has been studied extensively in African buffalo (*Syncerus caffer)*. Kruger National Park (KNP), South Africa represents a unique opportunity for studying the progression of BTB in a wildlife host. Historically, BTB was first introduced to the southern region of KNP in the 1960s from domestic cattle [1]. BTB was considered largely eradicated from the domestic animal populations surrounding KNP by the 1990s; however, it is thought to have persisted within KNP until 1990 undetected [1,3]. During the 1990s, prevalence of BTB within the KNP buffalo population steadily increased [3]. Previous research on herds of KNP buffalo have yielded a large data set complete with test conversion dates, gross pathology, and histopathology [13–16]. However, the progression of the disease within individual hosts has not yet been described. Understanding progression, including degree of necrosis in the lungs, proportion of lung grossly affected, and type of lymph nodes affected, in a free-ranging herd allows us to understand how BTB progresses in a natural system where animals have variable environmental conditions, food supply, and a broad range of coinfections [13–18]. This information is relevant to understanding human and bovine TB in similar variable conditions.

Previous research in the study herds identified regions of the African buffalo genome that associate with variation in host immune responses and infection resistance [13]. Two markers,

SNP 3195 and SNP 2253, were identified near genes involved in macrophage activation and pathogen degradation, increasing the risk of BTB by up to ninefold. Furthermore, animals carrying the risk allele at SNP 2253 marker displaced reduced activation of IL-12 from monocytes and macrophages; however, no significant pattern of cytokine production was found relative to SNP 3195 genotype. As such, the two SNP loci appear to associate with distinct mechanisms of infection resistance [13]. While these genetic regions have been investigated for disease resistance, their influence on disease progression has not been elucidated. Utilizing histology and gross pathology, as well as the previously identified SNP markers of interest, may help to further elucidate the evolutionary and ecological dynamics of BTB within this well studied model.

The goal of this study was to describe the progression of pathology associated with bovine tuberculosis over time in African buffalo within KNP. Relative pathology was quantified using number of lesions within lung lobes, level of necrosis, and pattern of lymph node infection. We then asked if variation among hosts in disease progression correlates to genetic markers of BTB risk previously identified within the study herds.

## Methods

### Ethics statement

Animal protocols for this study were approved by the University of Georgia (UGA) and Oregon State University (OSU) Institutional Animal Care and Use Committees (UGA AUP A2010 10-190-Y3-A5; OSU AUP 3822 and 4325).

### Study site

This study utilized data from a previously conducted longitudinal study performed from 2008 to 2012 [14]. A free-ranging population of African buffalo in Kruger National Park, South Africa, was followed to investigate the consequences of anthelmintic treatment on BTB dynamics [14]. Animals in the study were captured approximately every 180 days, over a 4-year period [14]. As part of the initial study, animals were separated into an experimental group that received a long-lasting anthelmintic, and a control group, which were untreated. At each subsequent capture, the buffalo cohort were tested for BTB using a whole-blood interferon gamma (IFNγ) assay (BOVIGAM, Prionics, Switzerland). This assay measures the difference in IFNy production of whole blood in response to incubation with mycobacterial antigens (PPD) [19]. For each animal, 2–9 BTB tests were obtained, depending on the time that each animal participated in the study (up to four years, 9 captures). For an animal to be considered positive for BTB, at least two consecutively positive BTB tests were needed [14]. Animals with inconclusive BTB test results were excluded from our analysis. Of those culled and necropsied, 38 animals had a known conversion date within 6 months (based on capture period), and fully completed pathological exams. For the SNP 3195 analysis, sample size included 36 buffalo, with 12 risk heterozygotes, 4 risk homozygotes, and 20 wildtype buffalo. For the SNP 2253 analysis, sample size included 36 buffalo with 11 risk heterozygotes, 1 risk homozygotes, and 24 wildtype buffalo. Animal protocols for this study were approved by the University of Georgia (UGA) and Oregon State University (OSU) Institutional Animal Care and Use Committees (UGA AUP A2010 10-190-Y3-A5; OSU AUP 3822 and 4325).

### BTB Progression–Gross and histopathologic examination

To quantify BTB progression, gross and histopathological examinations on culled animals were performed. South African National Parks Veterinary Wildlife Services (P. Buss) and

South African State Veterinary Service veterinarians (LM DeKlerk-Lorist, L.van Schalkwyk) followed standard protocols for culling and quantifying BTB progression during necropsies [14]. We localized infection and pathology over time in terms of (1) distribution of lesions in lymph nodes and their level of necrosis and (2) the spread of lesions and pathology within the lungs at necropsy. While there is currently no standardized metric for disease severity post-mortem in BTB, severity of pathological lesions and progression to necrosis as well as gross lesion burden have been utilized in disease modeling to assess severity [20]. Length of infection was determined as time from initial positive BTB test to time of cull. The original study separated the buffalo into a control or an experimental group that received anthelmintic treatment, to study the effects of anthelmintic treatment and BTB susceptibility and progression [14,15]. Whether the animal was in the control or experimental group of anthelmintic treatment was accounted for in all analyses. Treatment with anthelmintic was found to have no effect on BTB progression in the lung, but did have an effect in the lymph nodes, thus treatment is a covariate in the following models [15]. We then utilized previously identified BTB susceptibility risk alleles to test whether these gene variants also accounted for any of the differences in lung pathology.

The distribution of macroscopic BTB lesions in the lungs, respiratory lymph nodes (caudal mediastinal and tracheobronchial) and cranial lymph nodes of the head (mandibular, parotid, palatine tonsil, and retropharyngeal) were assessed at time of necropsy [14]. Lungs were examined grossly, pathology was diagrammed, and BTB lesions were measured. Sections of lung and serial sections of respiratory lymph nodes were collected from all BTB positive animals, regardless of whether gross pathology was observed. From animals where BTB lesions were observed, the largest BTB lesion was section and collected. Serial sections of the respiratory lymph nodes were collected, both those grossly affected and not. All sections were kept in 10% neutral buffered formalin [14]. All fixed tissues were routinely processed for histopathology and examined by a board-certified veterinary pathologist (K. Sakamoto).

## Statistical analyses

**Lesions over time in lymph nodes.** Using the survival package in R [21], lymph node infection status at time of necropsy was compared to the length of infection, creating a Kaplan-Meier curve. Mandibular, parotid, tonsillar, mediastinal, retropharyngeal, and tracheobronchial lymph nodes were all assessed as either histologically affected by tuberculosis or not, and the probability of each lymph node being infected at monthly time points of infection length was determined using Cox proportional-hazard models.

## Lesions over time in lungs

R packages lmertest [22], lme4 [23], and the base R package were used to evaluate the progression of bovine tuberculosis in this longitudinal study. Infection length was defined as the time from initial test conversion determined by IFNγ assay to time of culling. Using a generalized linear model we asked whether infection length correlated to percentage of lungs that were histologically necrotic, percentage of lungs histologically showing pathology but not yet necrotic, number of gross lesions per lung lobe and number of lobes grossly affected with lesions. We included herd location and treatment as covariates to account for these potential sourcs of variation. Pearson correlation matrices were created comparing total number of lesions and total number of lobes with the presence of gross lesions. Using the survival package in R [21], lung pathological status at time of necropsy was compared to the length of infection, creating a Kaplan-Meier curve of the probability the lung was affected at monthly time points of infection length using cox proportional-hazard methods.

### Genetic Basis of BTB Progression

R packages lmertest [22] and lme4 [23] were used to evaluate the potential effects of the risk alleles, SNP 3195 and SNP 2253 on BTB progression, simple linear regressions comparing the length of infection percentage of lungs that were histologically necrotic, percentage of lungs histologically showing pathology but not yet necrotic, number of gross lesions per lung lobe and number of lobes grossly affected with lesions were completed for each genotype (SNP 3195 wildtype, SNP 3195 heterozygote, and SNP 3195 homozygote, as well as for SNP 2253). General linear models utilizing the residuals from the above simple linear regressions were completed using the fixed variables of herd, whether the animal was a control or experimental treatment buffalo, and its risk allele genotype. GLM analyses was completed three times, once with each genotype (wild type, homozygote, or heterozygote) as the reference variable, running each genotype against the same variables (length of infection to percent of lobe histologically affected, number of gross lesions per lung lobe, and number of lobes with the presence of gross lesions). By running each genotype as the reference, we are able to compare how the genotypes differ with each variable.

## Results

### Summary statistics

A total of 312 buffalo were included over the course of the 4-year cohort study, with 137 animals culled at the end. Of these 137 buffalo, buffalo with incomplete data sets (i.e. missing histopathological data) or negative TB tests were removed, leaving 38 buffalo with sufficient data to be included in this study. The length of BTB infection ranged from 1 month to 42 months. The average duration of infection was 13 months, and the median length of infection was 7 months, including those infected up to 42 months only (animals infected over 42 months were positive at initial capture, thus we are unable to determine the length of infection).

### Lesion localization over time

BTB localized first in the lung, retropharyngeal, and tracheobronchial lymph nodes, before invading the lymph nodes of the head (tonsil, mandibular, and parotid)- consistent with infection via respiratory transmission (Fig 1). In the first year post-infection, most (80%) buffalo hosts did not show gross BTB lesions in any of the lymph nodes we examined or the lung. In animals that did have early BTB lesions, these were confined to the retropharyngeal lymph node. During the second year post-infection and first half of the third year (i.e., 13–30 months), these patterns remained relatively stable, with steady but incremental increases in the likelihood of gross lesions in the lung, tracheobronchial, and retropharyngeal lymph nodes, and lesions spreading only rarely to mediastinal, mandibular and parotid lymph nodes. By contrast, between 31–36 months post-infection, we observed a dramatic increase in the likelihood of pathological changes across the lungs and most lymph nodes: by 36 months post infection, >80% of buffalo had gross lesions in the lungs, and >40% in the tracheobronchial, retropharyngeal, and mediastinal lymph nodes. Thus, in the majority of studied animals, BTB manifested only minimally in terms of gross lesions for the first 2 to 3 years post infection.

### Patterns of spread within the lung

To understand how BTB lesions spread within buffalo lungs over time, we evaluated the distribution of gross lesions among lung lobes, as well as the total number of lesions detected in each animal's lung. We found that these measures were highly correlated (0.94–1.0), as BTB lesions tended to spread between lobes over time, resulting in a steady increase in total

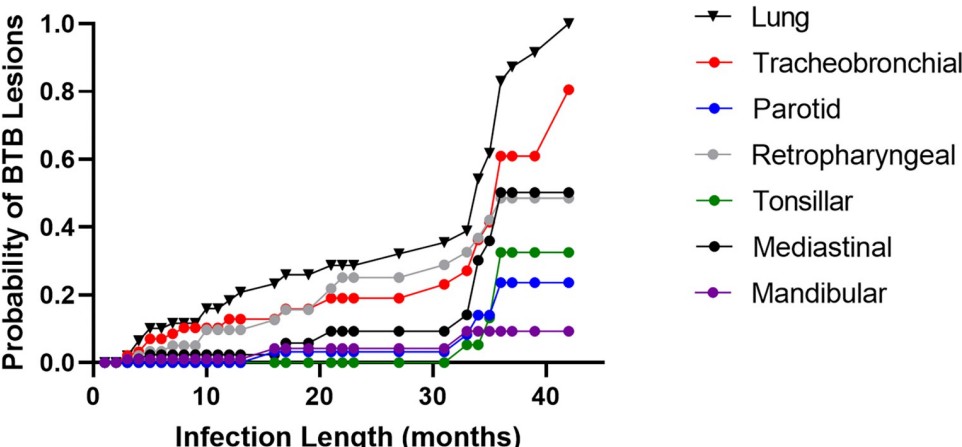

**Fig 1. BTB Infection Probabilities in Animals Testing Positive for BTB: Kaplan Meier curve showing the probability of infection in each lymph node and the lungs over time.**

detectable lesions overall (Fig 2A; statistical output is from the linear model, but for ease of visualization and since treatment was not statistically associated with the outcomes, we show the raw data and lines from a simple linear regression). Therefore, as the infection progressed, there was an increase in the number of lesions within a lobe, as well as the total number of lesions, indicating that the infection did not remain isolated to a single lobe, but spreads throughout the lungs as the number of lesions increases. The variance was high, however, for both number of lobes affected and number of lesions per all lobes, reflecting variability in the host-pathogen interaction among buffalo, and making it challenging to define a clear timeline for disease progression.

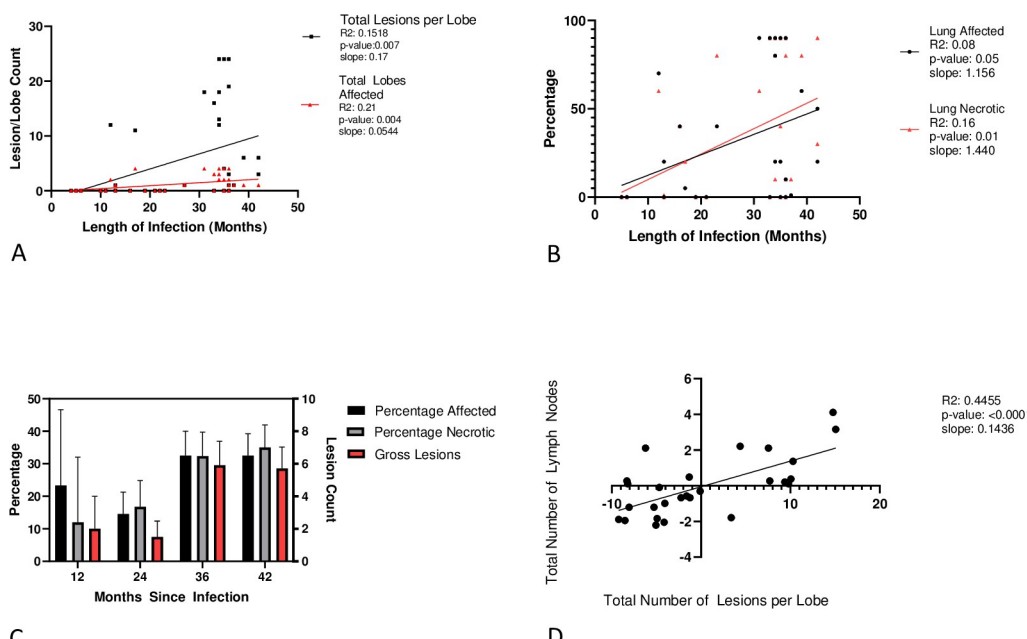

**Fig 2. Gross and Histopathological Progression of Bovine Tuberculosis A.** Gross Lesions within the Lungs: Simple linear regression showing the correlation between number of lesions in all lung lobes and number of lobes grossly affect with length of infection.

The percentage of lung that was necrotic as well as the percentage that was affected but not yet necrotic both increased with time since infection; percentage affected and necrotic was significant (Fig 2B, $p = 0.05$ and $p = 0.01$ respectively). As with the gross pathology, the variance was high, reinforcing the challenge of defining a clear timeline for disease progression. Also mirroring the gross pathology, at 36 months, there was an abrupt increase in percentages of the lung with histological lesions and presence of necrotic lesions (Fig 2C). **B.** Lung Histology: Simple linear regression showing the correlation between percent lung affected histologically and the length of infection, and the correlation between percent lung necrotic histologically and the length of infection **C**. Lung Pathology Averages: Bar graph showing the averages and SEM of the percentage of lung histologically affected, histologically necrotic, and number of gross lesions at specific time points. Within individual hosts, rapid development of lesions in the lungs was mirrored by rapid increase of gross lesions in the lymph nodes, consequently, it was unlikely to see high numbers of lesions per lung lobe with a low number of lymph nodes affected (Fig 2D). **D.** Gross Lesions within Lungs and Lymph Nodes Infected: Simple linear regression showing the correlation between the total number of lymph nodes infected and the total number of lesions per lobe at the same points of infection length.

## Genetic basis for histological progression

Previous work identified two genetic loci, SNP3195 and SNP 2253, which have been associated with an increased risk of acquiring *M. bovis* infection, compared to the wildtype alleles [13]. Here we tested whether these known risk alleles have any effect on progression of disease as well. SNP 2253 had no detectable effect on any of the measures of lung disease progression we examined. Herd and treatment were also tested and not significant. However, by all progression variables investigated, heterozygotes trend higher- increased number of lung lesions, number of lobes affected and percent of lesions microscopically necrotic, while the lesions in the homozygote trend less affected (decreased percentage of necrotic lesions and decreased number of lung lesions and lung lobes affected) based on expected progression of infection (Fig 3A and Table 1). When comparing the heterozygote to the wildtype buffaloes using linear regression, the heterozygote was disadvantaged both by the greater degree of histologically necrotic lung tissue early in the disease, and a slightly increased rate of necrosis compared to the wildtype (Fig 3B).

## Discussion

This study describes how BTB progresses in naturally infected African buffalo, which is very similar to described progression in cattle and humans, demonstrating their possible utility as a natural model system for the study of BTB and other mycobacterial infections.

BTB progresses at a slow to moderate pace initially in buffalo compared to cattle studies. A majority of animals had no gross lesions in year one, and steady but incremental progression though the next 18 months. Approximately 3.5 years post infection there was a striking increase in number of gross lesions and broader distribution of lesions in different tissues. Histological findings mirror this pattern, with a noticeable increase in lung tissue lesions at 36 months. The major difference between the progression in a wild buffalo model and controlled cattle studies is the speed of progression. In controlled studies, the speed of progression and severity of disease is proportional to the infectious dose administered and impacted by the route of inoculation [8]. Higher doses are often used to increase the speed of progression, to allow for experimental terminal end points within the 3 to 18 month range [12,24,25], thus progression that can take days in cattle experiments, may take months in wild buffalo. Additionally, the route of administration in cattle experiments differ, and affect the speed of

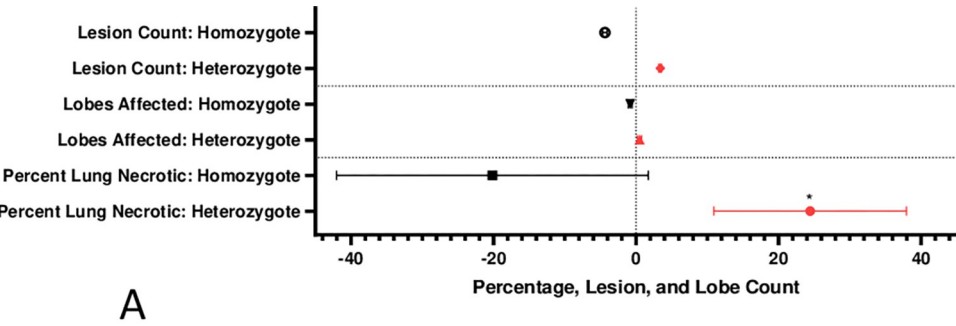

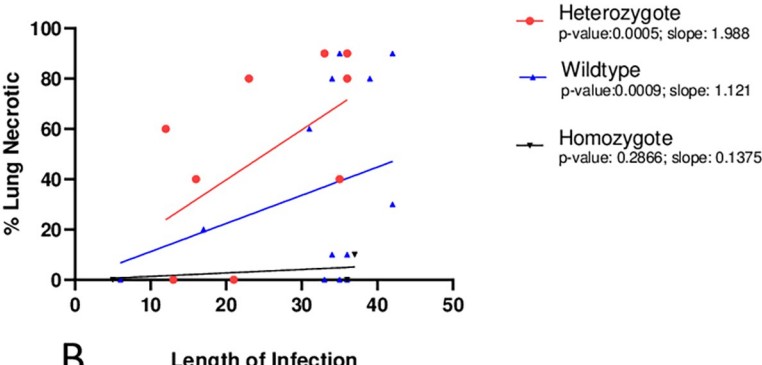

**Fig 3. Genotypic Variance of Pathological Progression A.** SNP 3195 Genotypes Compared to Wildtype: representation of general linear model values and SEM, showing the SNP3195 homozygote and the SNP3195 heterozygote compared to the wildtype in terms of lesion count within the lungs, lobes affected, and the percent of lung necrotic.*p-value = 0.0585 comparing homozygote and heterozygote percent lung necrotic.**B.** Percentage of Lung Necrotic by Genotype; SNP 3195: Simple linear regressions showing the correlation in the percentage of lung necrotic and the length of infection, accounting for genotype.

progression. Intratonsilar tends to lead to a more rapid initial disease spread, while intranasal and intratracheal lead to a slower progression, and are thought to resemble a natural infectious process more closely [24].

When comparing the pattern of spread, the buffalo show a progression of the disease through the lymph nodes similar to controlled cattle studies, however the cattle studies are at

**Table 1. Generalized linear models comparing the percentage of lung with necrotic lesions, number of lobes affected, and number of lesions within the lungs of the SNP 3195 homozygote (n = 4), heterozygotes (n = 12) and wildtype individuals (n = 20).** Positive values indicate metric was increased when compared to the wildtype individuals, negative values indicate metric was decreased compared to wildtype individuals.

|  |  | Percent of Lung Necrotic | Number of Lobes Affected | Number of Lesions |
|---|---|---|---|---|
| SNP 3195 Risk Heterozygote | Estimate Compared to Wildtype | 24.44 | 0.5001 | 3.391 |
|  | P-value | 0.0836 | 0.304 | 0.197 |
| SNP 3195 Risk Homozygote | Estimate Compared to Wildtype | -20.141 | -0.814 | -4.367 |
|  | P-value | 0.3665 | 0.272 | 0.274 |

an accelerated pace. A low dose ($1 \times 10^5$ colony forming units of *M. bovis*) inoculation revealed that retropharyngeal lymph nodes were microscopically and bacteriologically affected first, by two weeks post experimental inoculation, and grossly within one month [11]. Within one month post experimental inoculation, the tracheobronchial lymph nodes were microscopically affected, along with the mediastinal lymph nodes [11]. By 7 weeks, the lungs had gross lesions. Later, between 3 and 6 months, the parotid and mandibular lymph nodes, respectively, had pathological lesions [11]. In a high dose ($2 \times 10^7$ colony-forming units of *Mycobacterium bovis*) inoculation model, the same pattern, with a condensed time frame, was again demonstrated. Overall, by two weeks, the upper respiratory lymph nodes (defined as the retropharyngeal, parotids, and mandibular lymph nodes) all consistently had lesions in the cattle; by three weeks the bronchomediastinal lymph nodes were affected [12]. At the higher dose, there was higher variation between subjects in lung lobe distribution of lesions [12]. While the speed of progression remained dose dependent and multivariant, the pattern of progression remained consistent between controlled cattle studies and the African buffalo herd studied.

Rate of BTB progression was highly variable in African buffalo, as well as in cattle and humans [6,7], and the differences are most likely multifactorial [8,11,12]. Furthermore, BTB in wild buffalo appeared to parallel the natural cattle model of foci in the lymph nodes associated with the lung, followed by the lung becoming affected, more similarly to natural infection in humans [8,11] than what has been observed in controlled laboratory animal studies [8]. As the result of dynamic changes occurring in *M. bovis* infection, granuloma formation, necrosis, liquefaction, and eventual cavity rupture associated with each lesion behaves as an autonomous microenvironment undergoing its own progression independent of adjacent lesions and lobes, as well as different sites of infection [8,11]. The state of overall progression, therefore, is the sum of these local host-pathogen interactions [6].

In addition, the independent lesion developmental pattern of BTB has been proposed as a model of infection for *M. tuberculosis* infection in humans due to the similarities both clinically and immunologically [8]. Few long-term studies are available in cattle; however, variation in immune status is a known factor in the progression of BTB [6,25]. In mice, the commonly used model for human tuberculosis, there is less individual variation in susceptibility, with nearly all subjects infected succumbing to disease regardless of dose [8], as opposed to natural infections in humans and cattle, where roughly 30% of those exposed are infected [8]. In cattle and humans, the initial sites of infection remain consistent, however in murine models, the initial acute phase and infection takes place in the spleen and liver, not in the lymph nodes and lung [8]. Furthermore, the granulomas formed in cattle and humans are highly organized structures as described above, while in mice, a less organized, homogenous lesion with minimal necrosis is formed [8]. Ideally, parallel long term experiments comparing African buffalo to cattle studies could be performed, thus helping determine species versus context differences. Our results indicate that the African buffalo, a species well studied as a maintenance host for BTB, could also be utilized as a comparative model for human TB, with the potential for understanding progression in natural and variable environmental conditions over a longer interval.

The impact of host genetic variation on the susceptibility to infection, and the ability to control disease progression and pathogen replication, underpins the evolutionary response of host populations to novel pathogens [13]. Two genetic loci were associated with age of onset of BTB infection within the study herds, both suggesting a genetic component to susceptibility [13]. When assessing whether these previously identified genomic regions were associated with progression of disease, we found seemingly contradictory results for one region (SNP 3195). That is, homozygous animals tended to have a decrease in progression based on expected progression and percent of lung necrotic and number of lung lesions and lung lobes affected, while

wildtype and heterozygote animals tended to have an increase in these aspects of disease when compared to expected disease progression. Buffalo with one copy of risk allele at the SNP 3195 locus tended to progress faster, especially in terms of lung necrosis. The SNP 3195 heterozygote genotype tends to be associated with more necrotic lesions than the SNP 3195 homozygote genotype, at similar time points. The differences in proliferation of bacteria and subsequent necrosis within the lungs may be due to many mechanisms and interactions within the host, including but not limited to variation in the antigen specific immune response associated with BTB and the ability to balance appropriate inflammatory responses with tissue destruction. It is plausible that the SNP 3195 risk heterozygote, while having an increased degree of necrosis, remains prevalent in the buffalo population due to another benefit this genotype provides.

In conclusion, in naturally infected free-ranging buffalo, BTB progression followed a pattern similar to those seen in controlled experimental studies in cattle; in both study systems, the initial and most common sites of infection were the retropharyngeal and tracheobronchial lymph nodes and the lungs [6,8,11,12]. In the wild buffalo herds, we observed variability in BTB progression among animals, potentially due to differences in immune function and the exposure doses of *Mycobacterium*. However, uniquely, the buffalo population we studied showed a distinct increase in BTB progression at 36 months, which may be due to a wide variety of factors, such as repeat inoculation throughout the animals lifetime or changes in immune status, such as occur in pregnancy or lactation. Few long-term studies are available in cattle, in part due to their high experimental inoculation dose which result in rapid, acute disease progression [6,8], but does not capture the variation in exposure dose, a known factor in the progression of BTB [6,25] and human TB [8]. Long term studies are needed to assess the comparison between cattle and wild buffalo, as well as investigate the underlying immune mechanisms of disease progression and the animal's ability to go into a latency resembling human latency [8,25]. Lastly, there is evidence for a genetic component underlying the variability in pathology, specifically in the level of lung necrosis. There are known risk alleles for BTB susceptibility in this buffalo population; further work will be needed to discover novel loci affecting progression, including GWAS, aligning variation in BTB progression with genomic variation more broadly. Ultimately, BTB progression in naturally infected wild African buffalo in an uncontrolled environment mirrored the pattern but not the timescale of cattle studies, despite the plethora of natural variation in host immunity, nutrition, coinfections, and dose of exposure to mycobacteria. As a highly studied wildlife system, African buffalo could serve as a model for both cattle and, potentially, human tuberculosis, especially in the context of host-pathogen interaction in a more complex environment.

## Acknowledgments

We thank South African National Parks (SANParks) for permission to conduct this study in Kruger and M. Hofmeyr, P. Buss, and the entire SANParks Veterinary Wildlife Services Department for invaluable assistance with animal captures and project logistics. We thank R. Spaan, J. Spaan, K. Thompson, B. Beechler, P. Snyder, and B. Henrichs for work on animal captures and sample processing; A. Park for illuminating discussion about $R_0$ estimation; and S. Altizer and the Ezenwa and Altizer lab groups for valuable comments on earlier drafts of the manuscript.

## Author Contributions

**Conceptualization:** Hilary Ann Lakin, Vanessa O. Ezenwa, Brianna Beechler, Anna Jolles.

**Data curation:** Anna Jolles.

**Formal analysis:** Hilary Ann Lakin, Hannah Tavalire, Brianna Beechler, Anna Jolles.

**Investigation:** Kaori Sakamoto, Vanessa O. Ezenwa, Brianna Beechler, Anna Jolles.

**Methodology:** Peter Buss, Michele Miller, Sarah A. Budischak, Anna Jolles.

**Software:** Brianna Beechler.

**Supervision:** Brianna Beechler, Anna Jolles.

**Validation:** Brianna Beechler, Anna Jolles.

**Writing – original draft:** Hilary Ann Lakin.

**Writing – review & editing:** Hilary Ann Lakin, Hannah Tavalire, Kaori Sakamoto, Peter Buss, Michele Miller, Sarah A. Budischak, Kristina Raum, Vanessa O. Ezenwa, Brianna Beechler, Anna Jolles.

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
