## [Decision Letter · Decision Letter 0]

21 Sep 2022

Dear Dr. Hilary Ann Lakin,

Thank you very much for submitting your manuscript "Bovine tuberculosis in African buffalo (Syncerus caffer): progression of pathology during infection" for consideration at PLOS Neglected Tropical Diseases. As with all papers reviewed by the journal, your manuscript was reviewed by members of the editorial board and by several independent reviewers. The reviewers appreciated the attention to an important topic. Based on the reviews, we are likely to accept this manuscript for publication, providing that you modify the manuscript according to the review recommendations. 

Sincerely,

Jian-Wei Shao

Guest Editor

Elsio Wunder Jr

Section Editor

Reviewer's Responses to Questions

**Key Review Criteria Required for Acceptance?**

**Methods**

-Are the objectives of the study clearly articulated with a clear testable hypothesis stated?

-Is the study design appropriate to address the stated objectives?

-Is the population clearly described and appropriate for the hypothesis being tested?

-Is the sample size sufficient to ensure adequate power to address the hypothesis being tested?

-Were correct statistical analysis used to support conclusions?

-Are there concerns about ethical or regulatory requirements being met?

Reviewer #1: The authors tried to compare the distribution and progression of BTB lesion in buffalo with those of cattle. The progression of lesion in buffalo was investigated logitudinally. The objective of the study was clear. However, although it is possible to compare the types of tissues (lymph nodes) affected in the two species without running parallel experiments, it may not be possible to compare the progression of the infections between the two species without running parallell experimental infections under the same environmental conditions. Therefore, the authors cannot say much on the comparison of the progress of the infection in the two breeds. 

Although the authors indicated that 38 buffalos were able to be investigated for pathological studies, the information given on the study subjects is not sufficient to understand detail of study subjects. Regarding the hypothesis, the authors argue that buffalo can serve as model for the investigation of TB in cattle and in humans. This arquement is questionable or may be limited to buffalo raising countries if it works. Regarding the sample size, although the sample size of bovines used for comparison with the lesions of the 38 buffalos has not been described in the manuscript, the 38 buffalos used for this study are sufficient to generate data on the distribution and progression of lesion of BTB. However, the pathology examination procedures described in the method section were not supported with references. As far as statistical analysis is concerrned, with my limited knowldge of statistics, I felt that the authors used appropriate statistical analysis. Furthermore, the authors obtained approval from the ethical and regulatory bodies in order to undertake this study.

Reviewer #2: The objectives are clearly stated although the hypothesis is not stated as such. 

The study design is appropriate and the population clearly described.

The correct statistical analyses appear to have been used.

There are not ethical or regulatory concerns.

Lines 160-161: Why was length of infection determined from the initial positive IGRA result and to time of cull and subsequent histopathological examination? Shouldn’t the length of infection been from positive IGRA to euthanasia? Why was histopathological examination used in this determination?

Lines 169-170: mandibular, parotid, palatine tonsil and retropharyngeal lymph nodes are generally considered cranial lymph nodes of the head rather than respiratory lymph nodes as stated by the authors. 

Line 173: what was done with the tissues collected where gross lesions were not seen?

Lines 22-234: In assessing the invasion of various tissues why was bacteriological culture not used? 

Lines 159-160: The authors don’t justify why the metrics of lesion distribution, level of necrosis and extent of lesions in the lungs were used as measures of disease severity.

**Results**

-Does the analysis presented match the analysis plan?

-Are the results clearly and completely presented?

-Are the figures (Tables, Images) of sufficient quality for clarity?

Reviewer #1: The results describes only the progrssion and distribution of the pathology of BTB in buffalos. But there are no data on pathology of BTB in bovine. The tissues and lymph nodes affected were described. The progress of the lesion was influenced with time. The figures are of poor quality. Their visibility should be improved. The Tables do not qalify the standard of Tables of a scientific paper and hence should be modified.

Reviewer #2: The results are generally clearly stated.

Lines 255-256: what is meant by “rapid growth in the lymph nodes”? As written, is sounds as if the size of the lymph node was measured or bacterial burdens within lymph nodes were assessed.

**Conclusions**

-Are the conclusions supported by the data presented?

-Are the limitations of analysis clearly described?

-Do the authors discuss how these data can be helpful to advance our understanding of the topic under study?

-Is public health relevance addressed?

Reviewer #1: The authors seem to describe their own opinions rather than focusing of their findings. The authors did not include the limitation of the study in their manuscript. While they have forwared their opinions on how the result of this study can be used in the future and also described the the relevance of this study to the public health.

Reviewer #2: The conclusions are generally supported by the results and the limitations of the study are discussed. 

Lines 291-305: The authors focus on the differences in dosages as a means of accelerated spread of disease in cattle compared to the buffalo of the present study. It should also be mentioned that the routes of infection used by the references cited were different (intranasal vs intratonsilar), which may also affect disease progression.

**Editorial and Data Presentation Modifications?**

Reviewer #1: The manuscript requires editorial work. The order of citations in the body of the manuscript should be corrected. For example, reference 6 should appear before reference 3. The use abbrevations should follow the usual way of using abbrevation. The quality and visibility of the figures should be improved. The Tables should fulfill the standard of Tables of a scientific paper.

Reviewer #2: (No Response)

**Summary and General Comments**

Reviewer #1: The study was conducted on the distribution and progression of BTB in buffalos. The data were generated from 38 buffalos. The authors try to compare the lesions in buffalo with the lesions in bovine. According to the authors the distribtions of the lesions in the two species are similar. Furthermore, the authors claim that the progression of lesion is faster in bovine than in buffalo although their study could not address this question. In summary, the progression and distribution of lesion of BTB in buffalo can be published.

Reviewer #2: This manuscript describes a 4-year project to examine the progression of Mycobacterium bovis infection in African buffalo. The authors monitor infection status through semiannual interferon gamma release assays (IGRA), using 2 positive sequential IGRAs as the rough start date of infection. Thorough postmortem and microscopic examinations were done to evaluate the severity and extent of disease. Additionally, the authors compared their results on disease severity to the presence of 2 different SNPS associated with disease progression.

The manuscript is generally well written and represents a great amount of work with a difficult species with which to work. The authors compare and contrast disease progression in African buffalo to published results from experimental infection studies in cattle, recognizing that the African buffalo are exposed to a broad range of environmental challenges not present in experimental settings. Moreover, most experimental challenge studies in cattle do not last as long as the present study in buffalo due to the high cost of biocontainment.

PLOS authors have the option to publish the peer review history of their article (what does this mean?). If published, this will include your full peer review and any attached files.

Reviewer #1: Yes: Gobena Ameni

Reviewer #2: No

Figure Files:

Data Requirements:

Reproducibility:

References

---

## [Decision Letter · Decision Letter 1]

21 Oct 2022

Dear Ms Lakin,

We are pleased to inform you that your manuscript 'Bovine tuberculosis in African buffalo (Syncerus caffer): progression of pathology during infection' has been provisionally accepted for publication in PLOS Neglected Tropical Diseases.

Best regards,

Jian-Wei Shao

Guest Editor

Elsio Wunder Jr

Section Editor

Reviewer's Responses to Questions

**Key Review Criteria Required for Acceptance?**

**Methods**

-Are the objectives of the study clearly articulated with a clear testable hypothesis stated?

-Is the study design appropriate to address the stated objectives?

-Is the population clearly described and appropriate for the hypothesis being tested?

-Is the sample size sufficient to ensure adequate power to address the hypothesis being tested?

-Were correct statistical analysis used to support conclusions?

-Are there concerns about ethical or regulatory requirements being met?

Reviewer #1: Yes, my previous comments and concerns have been addressed.

Reviewer #2: The authors have addressed any concerns I had over the methods.

**Results**

-Does the analysis presented match the analysis plan?

-Are the results clearly and completely presented?

-Are the figures (Tables, Images) of sufficient quality for clarity?

Reviewer #1: Yes, my comments have been addressed.

Reviewer #2: The authors have satisfactorily addressed any concerns I had about the results.

**Conclusions**

-Are the conclusions supported by the data presented?

-Are the limitations of analysis clearly described?

-Do the authors discuss how these data can be helpful to advance our understanding of the topic under study?

-Is public health relevance addressed?

Reviewer #1: Yes, the conclusion has been revised and my earlier comments have been considered.

Reviewer #2: The authors have satisfactorily addressed any concerns I had about the conclusions.

**Editorial and Data Presentation Modifications?**

Reviewer #1: Accept

Reviewer #2: (No Response)

**Summary and General Comments**

Reviewer #1: This has been modified on the basis of my earlier comments.

Reviewer #2: The authors have satisfactorily addressed all my concerns.

PLOS authors have the option to publish the peer review history of their article (what does this mean?). If published, this will include your full peer review and any attached files.

Reviewer #1: **Yes: **Gobena Ameni

Reviewer #2: No

---

## [Editor Report · Acceptance letter]

8 Nov 2022

Dear Ms Lakin,

We are delighted to inform you that your manuscript, "Bovine tuberculosis in African buffalo (Syncerus caffer): progression of pathology during infection," has been formally accepted for publication in PLOS Neglected Tropical Diseases.

Best regards,

Shaden Kamhawi

co-Editor-in-Chief

Paul Brindley

co-Editor-in-Chief
